# Ultra-Fast Impedimetric Immunoassay for Detection of *Streptococcus agalactiae* Using Carbon Electrode with Nanodiamonds Film

**DOI:** 10.3390/mi14051076

**Published:** 2023-05-19

**Authors:** Daniel Bigus, Wioleta Lewandowska, Ewelina Bięga, Anna Grela, Aleksandra Siedlar, Marta Sosnowska, Magdalena Fabisiak, Tomasz Łęga, Yanina Dashkievich, Joanna Nowacka-Dośpiał, Katarzyna Palka, Sabina Żołędowska, Dawid Nidzworski

**Affiliations:** Institute of Biotechnology and Molecular Medicine, Kampinoska 25, 80-180 Gdansk, Poland

**Keywords:** nanodiamonds, potato starch, glassy carbon electrode, streptococcus agalactiae, immunosensor, immobilization, electrochemistry

## Abstract

This publication presents the results of work on the development of a quick and cheap electrochemical immunosensor for the diagnosis of infections with the pathogen *Streptococcus agalactiae*. The research was carried out on the basis of the modification of the well-known glassy carbon (GC) electrodes. The surface of the GC (glassy carbon) electrode was covered with a film made of nanodiamonds, which increased the number of sites for the attachment of anti-*Streptococcus agalactiae* antibodies. The GC surface was activated with EDC/NHS (1-Ethyl-3-(3-dimethylaminopropyl)carbodiimide/N-Hydroxysuccinimide). Determination of electrode characteristics after each modification step, performed using cyclic voltammetry (CV) and electrochemical impedance spectroscopy (EIS).

## 1. Introduction

*Streptococcus agalactiae* (also known as Group B Streptococcus or GBS) belongs to bacteria that are part of human microflora from the group of cocci. They inhabit the digestive system and the genitourinary tract. This streptococcus occurs in about 10–30% of healthy women and is usually a harmless bacterium that is naturally part of the female genital microbiota [1]. *Streptococcus agalactiae* is dangerous from the point of view of a woman of childbearing age and during pregnancy [2]. There are many changes in the reproductive organs of a pregnant woman, which significantly increase the risk of excessive growth of this bacterium. This is mainly due to the change in the pH vagina and hormonal changes (increase in estradiol) [3]. The infection may be mild and manifest with vaginal infections, or in severe cases, inflammation of the membranes, placenta or uterus may occur, which may result in premature birth, postpartum endometritis or even puerperal fever, which is a life-threatening condition [4]. Pregnant women colonized with GBS can transmit the bacteria to their newborns at the time of birth. The GBS colonization of newborns is caused by a vertical transmission through an oral cavity to a digestive and respiratory system and then by the bloodstream to distant organs. The risk of a newborn infection from the carrier mother is close to 70% [5]. Thus, GBS testing is especially important for pregnant women because these microorganisms may threaten the course of pregnancy and the newborn after its termination. *S. agalactiae* represents the main pathogen responsible for invasive infections and is the cause of sepsis and meningitis in newborns and young infants. In approximately 80% of cases, neonatal infection develops in the first week of life and is referred to as early-onset GBS disease. Late-onset GBS disease is defined as GBS infection in infants three months of age or older. The most common infections affecting these children are pneumonia, sepsis or meningitis [6,7]. Clinical and epidemiological studies have shown that intrapartum prophylaxis is the most effective way to reduce the risk of invasive GBS infection in the newborn. Widespread screening of pregnant women in the third trimester is crucial for detection of this pathogen. Finally, early diagnosis of GBS infection in pregnant women enables the introduction of antibiotic prophylaxis, which leads to a significant decrease in sepsis occurrence deliveries during the first seven days of babies’ lives [8]. 

There are many laboratory methods for the identification of group B streptococcus (Table 1). Due to the frequency of *Streptococcus agalactiae* most of the reference techniques used to detect the presence of microorganisms are based on the screening [9]. The microbial detection process is performed manually and requires a lot of work, which translates into a relatively long waiting time for the result, even up to 5 days. However, the major drawback of screening tests is the inability to identify women without risk factors and the introduction of antibiotic therapy in healthy women who have obtained false-positive results [10].

An alternative to screening methods is rapid laboratory tests for the specific detection of GBS. The most popular techniques are real-time qPCR [11,12], LAMP tests [13,14] and MALDI-TOF-MS [15,16]. Real-time PCR is the gold standard for sensitive, specific detection and quantification of nucleic acid that are diagnostic of, for example, infectious diseases, cancer and genetic abnormalities [17]. Another method for real-time detection is loop-mediated isothermal amplification (LAMP). LAMP is a very similar technique to qPCR, although it does not require a thermal cycler for the reaction [18]. Methods using MALDI-TOF MS are commercially available and validated. These systems rely on the detection of generic peptide patterns, limiting the discriminatory power for closely related species and the separation of subspecies [19]. However, despite their speed and widespread use, they have many disadvantages. The major disadvantage is the high cost of instruments, software and reagents. Another drawback is the need for qualified personnel and specialist knowledge of the techniques used. High requirements limit the use of these methods in many laboratories [20,21]. For all these reasons, it is important to create and develop fast, less costly tests for the specific detection of GBS, which can help in the early diagnosis of GBS infection.

The purpose of our study was to evaluate a rapid, reliable, easy-to-perform and inexpensive test to detect GBS that targets a surface immunogenic protein (Sip). GBS-specific Sip antigen is expressed by GBS strains of all serotypes [22].

The GBS-specific Sip antigen detection assay is based on the use of an electrochemical immunosensor based on the composite electrode, which consists of a modified carbon electrode with a nanodiamond-based film. The use of nanomaterials is already well-known as a powerful tool in the development of electrochemical sensors. Nanodiamonds (NDs) have found application in the modification of electrochemical sensors in view of their unique properties. The main advantages of NDs are high conductivity, mechanical resistance, chemical inertness and biocompatible surface, which can be easily modified [23,24]. Due to their properties, ND can potentially be used not only in medicine [25] but also in biology [26,27] and catalysis [28,29].

The functionalization of the sensor was characterized based on electrochemical techniques (Figure 1) such as electrochemical impedance spectroscopy (EIS) and cyclic voltammetry (CV). In this work, we describe the development of an immunosensor based on glassy carbon (GC) electrode modified by nanodiamonds. The developed immunosensor is based on the label-free electrochemical detection of the protein SIP (*S. agalactiae*). In this case, antibodies were attached to the electrode’s surface through a cross-linker. In the case of the GC electrode, carboxylic groups of antibodies activated by EDC/NHS react with amine groups of antibodies. The detection procedure is fast (in just 5 min), and no sample processing is required. These types of biosensors are able to detect the changes in electrochemical and/or electrical properties upon capture of the analyte.

Early diagnosis is essential to further reduce the incidence of early-onset GBS and to prevent the development of late-onset GBS infection. Moreover, easily available and simply performed intrapartum diagnosis of GBS infections may reduce the risk of complications in newborns.

**Table 1 micromachines-14-01076-t001:** A comparison of the analytical characteristics of the immunosensors developed in this work with relevant immunosensors for *S. agalactiae* detection based on the literature.

Type of Method	Detection Limit	Year	Reference
FISH	10^3^ and 10^4^ CFU/mL	2003	[30]
Rapid immunochromatographic test (ICT)	Range of 9.5 × 10^5^ to 3.7 × 10^6^ CFU/mL	2013	[31]
Amperometric	10 CFU/mL	2016	[32]
Multiplex PCR	2.8 × 10^4^ ng DNA	2017	[33]
LAMP	2.80 × 10^3^ genome copies mL^−1^	2017	[34]
Propidium monoazide–recombinase polymerase amplification (PMA-RPA)	1.2 × 10^3^ CFU/mL	2018	[35]
qPCR	10 copies/µL	2018	[36]
MALTI-TOF-MS	400 ppm	2019	[37]
real- LAMP	900 pg/μL	2019	[38]
Droplet digital PCR	5 pg/μL	2020	[20]
qPCR	1.68 fg/mL	2020	[39]
Multiple Cross Displacement Amplification Coupled With Lateral Flow Biosensor	300 fg/reaction	2020	[40]
The ratiometric LAMP electrochemical sensor	0.23 fg/μL	2020	[21]
CRISPR/Cas13-based assay	≈50 CFU/mL	2021	[41]
Fluorescent Impedimetric	6 CFU/mL	2021	[42]

## 2. Materials and Methods

### 2.1. Chemicals

Monoclonal antibodies (developed in mice) against Streptococcus agalactiae were obtained from ThermoFisher Scientific (Warsaw, Poland). The ND powder (monocrystalline Diamond Podwer (0–0.03 micron were obtained from Pureon (Lengwil, Switzerland). Potassium hexacyanoferrate (III), methanol, sodium nitrite and hydrochloric acid were purchased from Chempur (Piekary Śląskie, Poland). Phosphate-buffered saline (PBS), 4-aminobenzoic acid (4-ABA), N-hydroxysuccinimide (NHS), 1-ethyl-3-(3-dimethylaminopropyl)-carbodiimide (EDC), Tris-buffered saline (TBS) and sodium azide (NaN_3_) were obtained from Sigma–Aldrich (Poznan, Poland). Alumina slurries of 0.3 m were purchased from Buehler (Lake Bluff, IL, USA). Sulphuric acid, potassium hydroxide, hydrogen peroxide, ethanol and methanol were supplied by Pol-Aura (Warsaw, Poland). The potato starch (PS) was purchased at a local supermarket from Grula^®^ Trzemeszno, Poland).

### 2.2. Instrumentation

Potentiostat-galvanostat system (PalmSens4, Palmsens, Houten, The Netherlands), three-electrode assembly (Lambda System, Warsaw, Poland), glassy carbon disk electrode (3.0 mm in diameter, Mineral, Poland was used as a working electrode and Ag/AgCl/0.1 M KCl was used as a reference electrode, while Pt mesh served as an auxiliary electrode.

### 2.3. Biomaterials Preparation and Identification by Reference Method

#### 2.3.1. Recombinant Protein Production

Genes coding for recombinant proteins were chemically synthesized with codon optimization for *Escherichia coli* host expression (GenScript, Piscataway, NJ, USA). Synthetic genes were cloned into pET-51b(+) vector using *SalI* and *HindIII* restriction sites yielding N-terminal Strep-Tag II Fusion and C-terminal 10xHis-Tag fusion. Obtained plasmids were transformed into competent *Escherichia coli* BL21(DE3) (#C2527 New England Biolabs, Hitchin, UK). All recombinant proteins were induced in 1000 mL LB broth (#2020, A&A Biotechnology) using 0.1 mM Isopropyl β-D-thiogalactoside at OD_600_ = 0.5; 37 °C for 3 h with 180 rpm shaking. Cell cultures were centrifuged and pellets were lysed in 50 mL buffer containing: 50 mM NaH_2_PO_4_, 300 mM NaCl, 10% Triton X-100, 70,000 U/mL lysozyme (#62971 Merck, Darmstadt, Germany), Dnase I (#10104159001, Merck Germany) 15 µg/mL Ph = 8.0. Recombinant proteins were isolated from the lysate using IMAC chromatography with His-Select Nickel Affinity Gel (#P6611 Merck, Germany). Column gravity approach with 1 mL of resin has been performed (Table 2). 

#### 2.3.2. Preparation of ND-PS Dispersion and Sensor Fabrication

Potato starch dispersion was prepared in a proportion of 1.0 g of powdered potato starch to 100 mL of 5% acetic acid solution. The mixture was left under stirring at a temperature of 85 °C for 2 h until complete homogenization and a whitish transparent liquid was obtained. The resulting dispersion was stored under refrigeration. Then, 1.0 mg of NDs was added in 1.0 mL of MS, which remained in constant magnetic stirring for 2 h until complete dispersion homogenization. GCE was carefully polished with activated alumina, 1:1 proportion (*v*/*v*), for 5 min on a piece of clean cotton fabric and rinsed thoroughly with ultrapure water. Then, 5.0 μL of that dispersion was dropped on the GCE surface, and the solvent was evaporated at room temperature for 2 h. 

### 2.4. Immunosensor Fabrication

Prior to electrochemical measurements, glassy carbon electrodes with nanodiamonds film were cleaned using ethanol and demineralized water. Then, the electrodes were modified by polarizing the sample eleven times in a previously prepared deoxidized solution of diazonium salt [43]. 20 mg of 4-ABA was dissolved in 2 mL of 37% HCl (stirring for 15 min—average stirring speed 400 rpm). Then, it was cooled to 0 °C. Next, 2 mL of demineralized water was added to the mixture. The mixture was then stirred for a further 15 min to dissolve the precipitated 4-ABA chloride. Then 25 mg NaNO_2_ dissolved in 3 mL ddH2O was added dropwise for 30 min. After the addition of sodium nitrite, the compounds were stirred at 0 °C for about 10 min.

Modification of the GC surface was achieved by voltammetric electroreduction of the aryldiazonium reagents. The nitrosonium ion is formed in formation, which subsequently activates the amino group on 4-ABA. During CV sweeping, irreversible reduction peaks occur at a potential around 0.2 V. These peaks form due to the reduction of the diazonium precursor reagents by single electron transfer. Modification by electrode polarization from 0 V vs. Ag/AgCl to −1 V vs. Ag/AgCl five times at a speed of 100 mV/s was prepared with deoxidized diazonium salt solution with an Ag/AgCl (3M KCl) electrode as reference electrode (RE), and a platinum mesh as a counter electrode (CE). 

The samples were then washed with a strong stream of ddH2O and dried with a stream of argon, and 50 mM EDC and 100 mM NHS was placed on electrodes. This process lasted an hour and occurred at 4 °C. The samples were then washed with ddH2O, incubated with 10 µL of 0.1 µg/mL antibody solution and left for 24 h at 4 °C.

### 2.5. Electrochemical Measurements

The cyclic voltammetry and electrochemical impedance spectroscopy (EIS) were conducted using a Palmsens 4 potentiostat/galvanostat system (Methrom, Autolab, Utrecht, The Netherlands) in the standard three-electrode configuration. Glassy carbon electrode (Mineral, Poland) was used as a working electrode (GCE ∅ 3 mm), modified with a film from the dispersion of NDs in PS acid solution, Ag|AgCl (3.0 mol L^−1^ KCl) as a reference electrode; and wire of platinum as a counter electrode (Pt).

All the electrochemical tests were carried out in 5 mM K_3_[Fe(CN)_6_]/K_4_[Fe(CN)_6_] in 0.01 M PBS that was previously deaerated. In case of the electrochemical impedance spectroscopy measurements (EIS), the frequency ranged from 10 kHz to 1 Hz with 40 points. The amplitude of the AC signal was 10 mV. Each potential was held constant for 60 s before each measurement to obtain steady-state conditions. Obtained data were subjected to the analysis using EIS Spectrum Analyzer according to the proposed electric equivalent circuit (EEQC).

## 3. Results and Discussion

### 3.1. Electrochemical Characterization of Immunosensor

The electroactive area was estimated for GCE and GCE with nanodiamonds film, from Figure 2, using different scan rates (10 to 500 mV/s), in 5 mM K_3_[Fe(CN)_6_]/K_4_[Fe(CN)_6_] in 0.01 M PBS of an equimolar mixture of redox probe, respectively, using the Randles–Ševčík equation: I_p_ = 2.69 × 10^5^ AD^1/2^n^3/2^ ν^1/2^C
where I_p_ is the anodic or cathodic peak current, n is the number of electrons transferred (n = 1), A is the electroactive surface area, D is the diffusion coefficient (D = 7.6 × 10^−6^ cm^2^ s^−1^ for redox probe in 0.01 M PBS solution and C is the redox concentration.

Figure 2 shows a comparison of cyclic voltammograms between the GC and GC with NDs electrodes in the presence of 5 mM K_3_[Fe(CN)_6_]/K_4_[Fe(CN)_6_] in 0.01 M PBS solution at a scan rate of 10 mV/s–500 mV/s. The difference of peak potentials (ΔEp) for the GC with NDs electrode was 200 mV, while for the GCE, it was 243 mV, showing an improvement in reversibility for the redox pair for the GC electrode with nanodiamonds film. These results indicate that the GC with NDs exhibits remarkably better electrochemical performance than the GC. 

Checking the correctness of successive modifications of the electrode was carried out by CV and EIS. All electrochemical measurements made were using a solution buffer in PBS solution, pH 7.4, containing 5 mM K_3_[Fe(CN)_6_] and 5 mM K_4_[Fe(CN)_6_]. The applied redox system enables the analysis of changes in the kinetics of electron transfer on the surface, thanks to which we are able to notice changes after each modification stage, which allows us to determine the correctness of each step. CV measurements were performed in the potential ranges from −0.20 V to 0.60 V with a scan rate of 0.1 V/s. The obtained potential differences are related to the electrochemical window characteristic for each electrode. The obtained CV spectra provide information about changes in charge transfer, and EIS measurements inform about changes in resistance that occur on the electrode surface.

Data from the electrochemical measurements during carbon surface with Nanodiamonds modification are shown in Figure 3A. In Figure 3B, it is easy to follow the changes recorded during the successive steps of the sensor modification recorded by the CV. In addition to the previously discussed changes after incubation in Nanodiamonds, there is a clear decrease in the height of the current peaks from the oxidation and reduction in [Fe(CN)_6_] ^3−/4−^ by 16 μA.

However, when it is hard to see the binding receptor to the surface and saturating the free sites with BSA, it is much easier to interpret changes occurring on the sample from the impedance spectra shown in Figure 3B. To better present the data, the impedance spectra are shown in the range from 10 kHz to 1 Hz. To improve data analysis from the EIS, they were fitted to an equivalent electrical circuit (EEC), which is shown in Figure 3C, and the results are shown in Figure 3C with the chi-square parameter interpreted as a goodness of fit. 

### 3.2. Detection of Streptococcus Agalactiae Protein

In order to obtain a satisfactory level of resistance changes on the tested electrodes, the influence of the incubation time of the analyzed sample on the level of the obtained response was checked. For this purpose, analyzes were carried out in which the incubation time of the analyzed samples was as follows: 3 min, 5 min, 7 min and 10 min (Figure 4). 

The results obtained for the following times are presented in Table 3 below. 

The analysis of the obtained results indicated no significant differences in the level of received responses to the sample. However, it can be argued that the optimal range of incubation will be up to 5 min due to the fact that after this time, the level of response to the sample did not change or was lower. For this reason, the final incubation time for each of the positive and negative control samples analyzed was 5 min.

### 3.3. Biosensor Selectivity, Repeatability and Stability Studies

Mycoplasma hominis, Ureaplasma, Streptococcus agalactiae, Gardnerella vaginalis bacteria were used as potentially interfering bacteria to investigate the selectivity of the presented immunosensor. The bacteria concentration was kept in the same order of magnitude to receive comparable results. After 5 min incubation, the EIS spectra were recorded. According to Figure 5, all negative controls did not give a substantial impedance increase, the percentage change of Rct did8 not exceed 25% for both single samples, and this value was established as a threshold for the distinction between positive and negative samples. 

After a glassy carbon electrode with nanodiamonds film modification, it was placed in an electrochemical cell, and EIS spectra were recorded until system stabilization was observed. The stability of the sensor was verified by two additions of PBS to exclude unspecific interactions. In the case of the immunosensor, all pathogens sample dissolved in 0.01 M PBS was firstly incubated on the electrode surface for a given time, rinsed with PBS and immediately immersed in fresh 5 mM K_3_[Fe(CN)_6_]/K_4_[Fe(CN)_6_]/0.01 M PBS solution for EIS measurement. Figure 5 shows the impedance spectrum recorded during subsequent additions of the protein solution with increasing concentrations, which was preceded by two-times additions of its solvent (PBS) as a negative control Table 3 presents GC–based immunosensor response after incubation in the protein sample, expressed as charge transfer resistance change (ΔR_ct_). All R_ct_ change values were calculated from the equation:$$Sensorresponse=\frac{R^{Test}{}_{ct}-R^{Basic}{}_{ct}}{R^{Test}{}_{ct}}\times 100\left(\%\right),$$
where Rct Test is for the sample and Rct Basic is for the fully prepared immunosensor.

The EIS was used to investigate the metrological performance of the biosensor detecting the SARS-CoV-2 virus protein N by spotting the solutions with different concentrations (1.38 pg/mL, 13.8 pg/mL, 13.8 ng/mL, 0.138 ng/mL, 1.38 μg/mL) on the surface of electrodes and incubating them for optimal time (Figure 6). 

The limit of detection was calculated from the relation LOD = 3 × SD/slope, where SD is the standard deviation in the low concentration range. For all tested surfaces, we obtained a wide linear range of concentrations from 1.38 ug/mL to 1.38 pg/mL. 

## 4. Conclusions

This work presented the design and characterization of *Streptococcus agalactiae* antibodies immobilization onto a glassy carbon electrode with nanodiamonds film for impedimetric detection of *Streptococcus agalactiae* protein. The assay could detect *Streptococcus agalactiae* protein at concentrations as low as 1.38 pg/mL with a linear detection range of 2.92 ng/mL (R^2^ = 0.98). The advantage of this ‘one-step’ diagnostic assay relative to an ELISA or mass spectrometry is a rapid and sensitive measurement of antigen binding to nanodiamonds film. This provides a proof of concept that we intend to use to develop a clinical test with swabs from gynecological patients in order to determine whether *Streptococcus agalactiae* bacteria detection in patient fluid provides any prognostic indicator. Therefore, the electrode modified with NDs has shown very promising results in the electrochemical sensing of *Streptococcus agalactiae*. 

## Figures and Tables

**Figure 1 micromachines-14-01076-f001:**
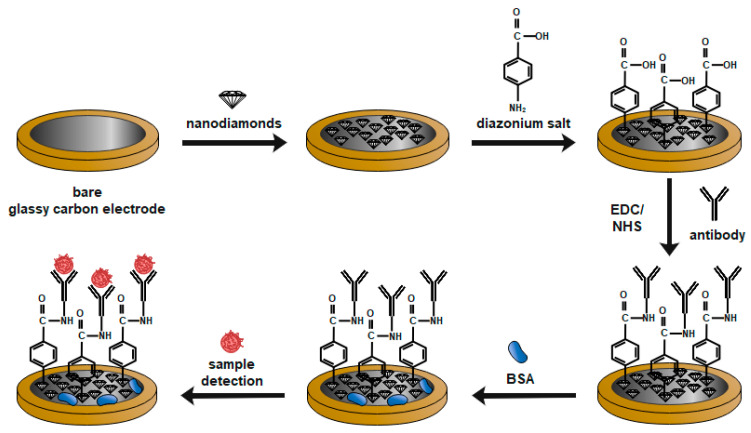
Mechanism of glassy carbon electrode with nanodiamonds film modification.

**Figure 2 micromachines-14-01076-f002:**
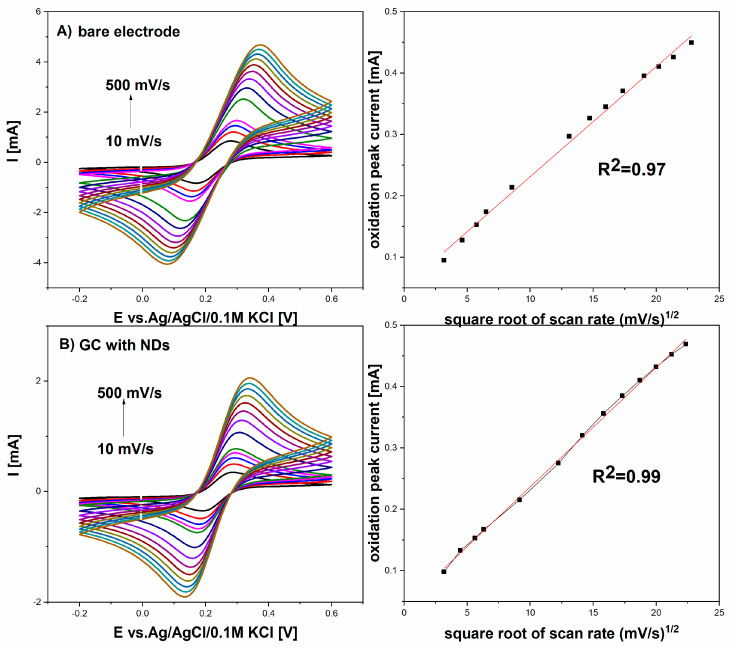
Cyclic voltammograms recorded at different scan rates (10–500 mV/s) using bare GC electrode and GC with NDs. (**A**) bare electrode (**B**) GC electrode with NDs.

**Figure 3 micromachines-14-01076-f003:**
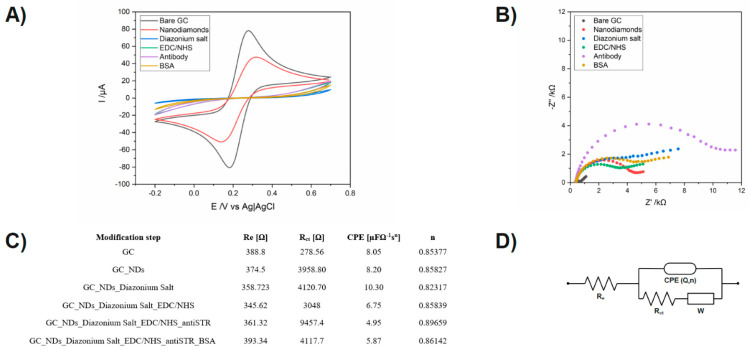
Measurements of biosensor fabrication steps: (**A**) cyclic voltammograms for the bare and modified GC electrode, (**B**) electrochemical impedance spectra for the bare and modified GC electrode, (**C**) List of values of elements calculated from the electric equivalent circuit (EEQC) for bare and modified electrodes. (**D**) electric equivalent circuit (EEQC) utilized for fitting and data analysis.

**Figure 4 micromachines-14-01076-f004:**
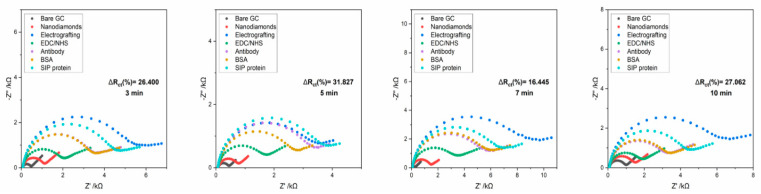
The impedance spectra registered for carbon electrode modified Nanodiamonds film with antibodies incubated for different periods (3–10 min) with *S. agalactiae* protein, registered in 5 mM K_3_[Fe(CN)_6_]/K_4_[Fe(CN)_6_]/0.01 M PBS.

**Figure 5 micromachines-14-01076-f005:**
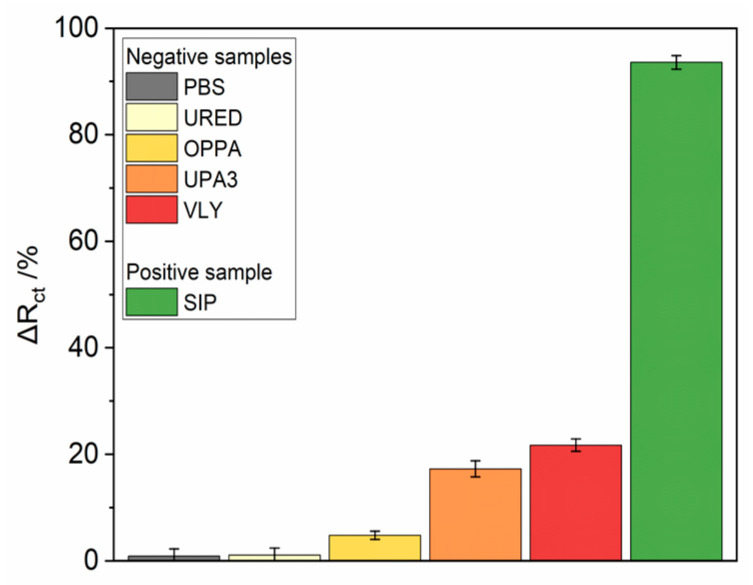
The selectivity of the immunosensor tested versus positive *S. agalactiae* sample and other pathogens: Ured, Oppa, Upa3, Vly, PBS bare buffer solution was shown as blank and reference sample.

**Figure 6 micromachines-14-01076-f006:**
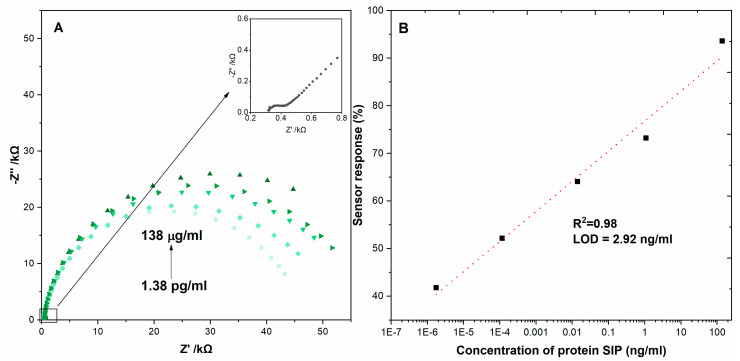
(**A**) Impedance spectra of immunosensor after incubating in solutions with different protein concentrations recorded in 5 mM K_3_Fe(CN)_6_/K_4_[Fe(CN)_6_]/0.01 M PBS. (**B**) The relation between the sensor response expressed as Rct change (ΔRct) and the protein concentration. Registered in 5 mM K_3_[Fe(CN)_6_]/K_4_[Fe(CN)_6_]/0.01 M PBS. Error bars denote confidence interval (α = 0.05, n = 3).

**Table 2 micromachines-14-01076-t002:** Recombinant Protein Imac Chromatography Conditions.

Chromatography Step	Upa3 Genebank: WP_006688445.1	UreD Genebank: AAF30840.1	Oppa GenBank: CAX37285.1	VLY Genebank: ACD39459.1	Sip Genebank: AAG18474.1
**Column Wash**	50 mM NaH_2_PO_4_, 300 mM NaCl, 50 mM Imidazole, pH = 8.0 20 CV	50 mM NaH_2_PO_4_, 300 mM NaCl, 10 mM imidazole, pH = 8.0 20 CV	50 mM NaH_2_PO_4_, 300 mM NaCl, 20 mM Imidazole, pH = 8.0 20 CV	50 mM NaH_2_PO_4_, 300 mM NaCl, 25 mM Imidazole, pH = 8.0 20 CV	50 mM NaH_2_PO_4_, 300 mM NaCl, 5 mM Imidazole, pH = 8.0 20 CV
**Elution**	50 mM NaH_2_PO_4_, 300 mM NaCl, 150 mM Imidazole, pH = 8.0 20 CV	50 mM NaH_2_PO_4_, 300 mM NaCl, 250 mM Imidazole, pH = 8.0 20 CV	50 mM NaH_2_PO_4_, 300 mM NaCl, 100 mM Imidazole, pH = 8.0 20 CV	50 mM NaH_2_PO_4_, 300 mM NaCl, 250 mM Imidazole, pH = 8.0 20 CV	50 mM NaH_2_PO_4_, 300 mM NaCl, 250 mM Imidazole, pH = 8.0 20 CV

**Table 3 micromachines-14-01076-t003:** Rct value of biosensor response in time towards *S. agalactiae* samples.

Incubation Time	∆R_ct_ (%)
3 min	26,400
5 min	31,827
7 min	16,445
10 min	27,065

## Data Availability

The data will be made available at a reasonable request to the corresponding author.

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
