# Peer review of "Ultra-Fast Impedimetric Immunoassay for Detection of Streptococcus agalactiae Using Carbon Electrode with Nanodiamonds Film"

_micromachines, 2023, doi:10.3390/mi14051076_

Round 1

Reviewer 1 Report

Though the study by Bigus et al. on “Ultra-fast impedimetric immunoassay for detection of Streptococcus agalactiae using carbon electrode with nanodiamonds film” is reasonable. However, the authors may consider doing necessary amendments to the manuscript for better comprehensibility of the study.

It would be interesting to know. the followings:

Could you please comment on the other recent methods, like fast fluorescent screening assay?

Could you please comment on your results with other methods, for example, using immunological markers of R4 (R4/Alp3) (or other GBS protein)?

 Or with 16S rRNA S. agalactiae genes by qPCR?

I found overlapping of the texts, please consider rewriting these portions

Line 78: Sip protein is exposed at the surface of intact GBS cells, where it is accessible to specific antibodies.

Lines 152-171

Lines 221-232

Lines 293-294

Lines 297-298

Reviewer 2 Report

Bigus et al. have reported an ultrafast detection of pathogen Streptococcus agalactiae in ultralow concentration using electrochemistry tools such as impedance and CV measurements. Creating new pathways for pathogen detection is a novel technique and have fundamental importance for future research also. In this regard the manuscript is well written and have enough data for support. But based on that I can not accept the manuscript in its present form. I hope authors will answer my queries and comments during revising the manuscript.

1)      Author claims in Figure 2 that there is a substantial shift of peak potential for GC electrode in presence of ND but the both the figure looks very identical to each other. Author should provide some characterization proof for binding of NDs on top of GC bare electrode.

2)      The data quality specially the resolution of Figure 2 should be improved.

3)      The author should comment on the long period of incubation time specially more than 30 mins.

4)      Can a microscopic picture of the modified film be addressed?

Round 2

Reviewer 1 Report

This revised version can be accepted.

Reviewer 2 Report

Manuscript has been revised properly in its current version. It can be accepted in present form.